# Analyzing cold hardiness (Based on DTA) of one-year-old branches of peaches

**Yonghong Li** [1], **Jie Li**[1], **Zhaoyuan Wang**[1], **Guojian Liu**[1], **Yu Wang**[1], **Ruifeng Chang**[1], **Hu Chen**[1], **Qihang Tian**[1]*, **Xiaodi Wang**[2]*

**1** Research Institute of Pomology of CAAS, Xingcheng, Liaoning, China, **2** Changli Fruit Institute, Hebei Academy of Agriculture and Forestry Sciences, Changli, Hebei, China

* liyonghongpeng@163.com (QT); wangxiaodi@caas.cn (XW)

**Data Availability Statement:** All relevant data are within the manuscript.

**Funding:** This work was supported by Key Laboratory of Biology and Genetic Improvement of Horticultural Crops (Germplasm Resources

## Abstract

In this study, we conducted a low-temperature exothermic (LTE) investigation on 1-year-old (1a) branches of sixteen peach cultivars through a differential thermal analysis (DTA) procedure. We used a three-point approach to determine the lethal injury temperature (LT-I) of the xylem, the LTE correlation indexes, and the subordinate function value method were applied to compare cold hardiness of sixteen peach varieties. The results showed that the slope of the LT-I for the xylem of sixteen peach cultivars was different, and the LTE indexes were significantly different. Among all the studied varieties, the cold hardiness was strongest in Donghe No.1, followed by Wangjiazhuangmaotao No.2 and Hunchun. Qiuyan and Yanhong are second, and belong to the cold-resistant type; Qiuyi, Okubo, Zhongnongjinhui, and Chunmei, exhibited medium cold hardiness. Zhongtaohongyu, Spring snow, Yufei, and Zhongyou No.8 varieties exhibited low hardiness; while the 21st century, Golden Honey No. 1 and Zhonghuashoutao have the worst cold hardiness and are the weakest cold-hardiness types. In addition, the injury degrees of xylem from LT-I analysis were significantly related to the browning rates (BR) and electrolytic leakage (EI) from traditional low temperature freezing analysis. It is demonstrated that the LTE analysis is a simple, accurate, and practical method for identifying the cold hardiness of 1a branches of peach.

## Introduction

Peach (*Prunus persica* (L.) Batsch), a nutrient-rich fruit, has been extensively distributed worldwide [1, 2]. However, its growth, development, and distribution are restricted by low temperature, a crucial constraint [3]. There are many reports (national and international) on the identification methods and cold hardiness physiology of peach. The methods of EI are used to compare the cold hardiness between varieties by combining the contents of physiological and biochemical indexes such as free water, MDA, soluble sugars, soluble proteins and proline, germination rate, BR, and recovery growth methods [4–7]. It is time-consuming, laborious, and has many influencing factors.

Since the 1970s, the low temperature exothermic (LTE) technique has been improved considerably, thus allowing the comparison of cold hardiness across various plant varieties. For

Utilization), Ministry of Agriculture,P.R.China (No. NYZS202305) and the Fundamental Research Funds for Hebei Academy of Agriculture and Forestry Sciences(No. 2023020102), Technology Innovation Special Project of Hebei Academy (No. 2022KJCXZX-CGS-6), Key R&D Projects of Hebei Province (No.21326310D), Modern Agricultural Industrial Technology System of Hebei Province (No. HBCT2023130205) and China Agriculture Research System of MOF and MARA (No. CARS-30-Z-02). The funders played the important role in study design, data collection and analysis, decision to publish, and preparation of the manuscript.

**Competing interests:** The authors received funding from Key Laboratory of Biology and Genetic Improvement of Horticultural Crops (Germplasm Resources Utilization), Ministry of Agriculture,P.R. China (No. NYZS202305) and the Fundamental Research Funds for Hebei Academy of Agriculture and Forestry Sciences(No. 2023020102). This does not alter their adherence to PLOS ONE policies on sharing data and materials. There are no patents, products in development, or marketed products associated with this research to declare.

LTE examination, recording, and analysis, in the LTE technique, ice crystals are formed in tissues at low temperatures with a PC-assisted procedure of differential thermal analysis (DTA), as well as further cold hardiness evaluation of plant tissues [8, 9]. In previous studies, data analysis was performed by the half-lethal temperature (LT50) or LT10 method [10–13], which was unable to sufficiently reflect the cold hardiness trait of the different varieties. Later, Gu et al. [14] performed a cold hardiness assessment of buds by using slope Qlt and LT50, which were more comprehensive and accurate, and provided a new way for analyzing the cold hardiness of branches. The comprehensive evaluation of plant cold hardiness has been performed effectively by the subordinate function method for poplar [15], grape [16], and apple [17].

Flower buds and trunk tissues are susceptible to freezing or frost damage [18]. Minas et al. [19] have revealed that DTA accurately predicts mid- and late-winter cold hardiness and damage of peach floral buds, however, it loses the capacity to determine ice nucleation events as bud development advances in spring. Currently, there is no report on the use of the LTE method to analyze the cold-hardiness of the 1a branches of peach varieties. In this study, using dormant branches of 16 peach varieties, the exothermic characteristics and LTE-related indexes of each variety were analyzed by the LTE method, and the regression analysis was performed for the xylem lethal injury temperature (LT-I). The cold-hardiness of 1a branches of 16 peach varieties was comprehensively evaluated by the membership function method combined with LTE-related indexes, and the results were verified by the conductivity method and the tissue browning method. To provide technical support for cold-hardiness identification of peach germplasm, cold hardiness breeding, and planting regionalization of existing varieties, an accurate and reliable method for cold hardiness identification of peach was established.

## Materials and methods

### Experimental materials

Trials were performed within the peach orchard in Changli, China (39°45′N, 119°12′E). It is an area with a continental monsoon type of climate, with four seasons. The *Prunus persica* cultivars used in this study included Wangjiazhuangmaotao No.2, Hunchun, Donghe No.1, Qiuyan, Yanhong, Qiuyi, Okubo, Zhongnongjinhui, Chunmei, Zhongtaohongyu, Spring snow, Yufei, Zhongyou No.8, 21st century, Golden honey No.1, and Zhonghuashoutao.

### Experimental instruments

The DTA system used in this study comprised a data acquiring unit (fabricated by the Mechanical and Electronic Engineering School, Shandong Agricultural University, China) and programmable temperature incubators (GDTS-2252, Wuxi Jingchuang Technology Co., Ltd, China). The incubators could be operated between –70 and +150°C and were 600 mm × 750 mm × 500 mm in size. The data acquiring unit included 27 Peltier thermoelectric modules (40 mm × 40 mm × 15 mm). This unit was responsible for sensing freezing exotherms and converting them into voltage outputs. Additionally, it could also record temperature variation during exothermic procedures through a pt100 sensor, with an accuracy reaching 0.1%.

### Experimental methods

**LTE determination method.** The 1a branches of the different varieties were cut with a pair of scissors into 2.0-cm-long sections that were 0.8–1.0 cm in diameter. The sections were each placed into individual thermoelectric modules (triplicates) and pressed using a plastic foam board (600 mm × 750 mm × 500 mm) to maintain adequate contact with the

thermoelectric modules. Later, the temperature was reduced as follows: 1 h at room temperature to 4°C, 1 h at 0°C, 11 h from 0°C to –44°C, and 1 h at –44°C [1, 13, 20, 21].

**Determination of EI and BR.** 'Wangjiazhuangmaotao No.2', 'zhongnongjinhui' and 'Chunmei' branches were collected to detect the electrolytic leakage (EI) and the browning rate (BR). On 5th January 2020, branches of equal diameter (0.8–1.0 cm) and length (25–30 cm) (from the same batch of samples as LTE) were harvested from all sides of the 7-year-old trees. The branches were cleaned with distilled water and separated into six groups, each of which was wrapped with an adequate amount of gauze and placed in a plastic bag. The temperature treatments provided were 4°C (control), –23°C, –27°C, –31°C, –35°C and –39°C, with 12 h of treatment time.

For EI estimation, the method used was as described in a previous study [22].

BR was determined by the tissue browning method [18]. Twenty branches of each variety were examined. From actual observations and the browning area of the secondary xylem, a five-grade classification system was adopted to classify the freezing injury. The categories were as follows: the browning area of Grade 0 xylem was 0–3%, Grade 1 was 4–30%, Grade 2 was 31–50%, Grade 3 was 51–75%, and Grade 4 was 76–100%.

## Data analysis

The SPSS 20.0 software (SPSS Inc., Chicago, IL, U.S.A.) was used for all statistical analyses; analysis of variance and correlation analyses were performed. All data are expressed as the mean ±standard error; the significance level was set at $P < 0.01$. The root cold-hardiness analysis was used to perform the LT-I regression [23]. The LSD approach was adopted to perform analysis of variance (ANOVA) for the lethal temperature coefficient (LT50) and the slope (Qlt), shown in Table 1. The severity of the injury of all cultivars was estimated based on the LT-I at various temperatures (Table 2). The subordinate function value analysis was performed for evaluating the cold hardiness of the 1a branches comprehensively using the formula:

$$R\,(Xi) = 1-(Xi-Xmin)/(Xmax-Xmin).$$

Here, R is the value of the subordinate function; Xi is the measured value of a certain index; Xmin and Xmax represent the minimum and maximum values, respectively, of the index in all resources tested. The value of the subordinate function of each LTE index was calculated by the above-mentioned formula, and the mean value of the LTE index of each variety was taken as the mean value of the subordinate degree. Based on the average subordinate degree, the cold hardiness was divided into five grades: 0.70–1.00 was high hardiness (HH), Grade 1; 0.60–0.69 was hardiness (H), Grade 2; 0.40–0.59 was medium hardiness (MH), Grade 3; 0.30–0.39 was low hardiness (LH), Grade 4; 0–0.29 was the weakest grade (WH), Grade 5.

## Results

### Construction of the LT-I regression equation and estimation of Qlt & LT50

The DTA system was used to obtain the LTE pattern of the peach branches. The continuous cooling process revealed two exothermic peaks, according to the exothermic characteristics. The first exothermic peak observed above –10°C was attributed to a high-temperature exothermic process (Fig 1). The second exothermic peak occurred between –25°C and –40°C (Fig 1), indicating the presence of LTE characteristics. The first exothermic process did not cause any freezing injury to the branches, indicating that the exothermic treatment did not cause freezing injury associated with the freezing of intercellular water and the xylem conduit gravity water.

**Table 1. LT-I equation of 1-year-old branch xylem of 16 peach varieties.**

| Variety | LT-I | $R^2$ | Qlt | LT50/˚C | LT0-LT100/˚C | The difference value between LT0 and LT100/˚C |
|---|---|---|---|---|---|---|
| Zhongtaohongyu | y = -7.65x - 187.05 | 0.967 | -7.65AB | -30.99B | -24.63–37.31 | 12.68 |
| Qiuyan | y = -10.57x - 294.43 | 0.991 | -10.57C | -32.59DEF | -27.90–37.28 | 9.38 |
| Hunchun | y = -11.27x - 324.23 | 0.995 | -11.27C | -33.21EF | -28.79–37.62 | 8.83 |
| 21st Century | y = -6.55x - 137.58 | 0.996 | -6.55A | -28.64A | -21.05–36.27 | 15.22 |
| Spring Snow | y = -7.41x - 179.87 | 0.995 | -7.41AB | -31.02B | -24.32–37.76 | 13.44 |
| Zhonghuashoutao | y = -6.47x - 146.30 | 0.964 | -6.47A | -30.34BC | -22.40–37.29 | 14.89 |
| Qiuyi | y = -7.85x - 196.83 | 0.990 | -7.85AB | -31.44BCD | -25.50–38.11 | 13.20 |
| Golden honey No.1 | y = -6.57x - 144.70 | 0.988 | -6.57AB | -29.63A | -22.02–37.25 | 15.23 |
| Okubo | y = -6.46x - 159.43 | 0.983 | -6.46A | -32.42BCD | -25.53–38.34 | 12.81 |
| Yanhong | y = -7.27x - 190.78 | 0.931 | -7.27AB | -33.12BCD | -27.53–40.34 | 12.81 |
| Yufei | y = -7.99x -197.66 | 0.989 | -7.99B | -31.00B | -24.81–37.19 | 12.38 |
| Wangjiazhuangmaotao No.2 | y = -12.31x - 370.48 | 0.974 | -12.31D | -34.16EF | -30.54–38.46 | 7.92 |
| Zhongyoutao No.8 | y = -7.59x - 185.44 | 0.943 | -7.59B | -31.02B | -24.81–37.22 | 12.41 |
| Donghe No.1 | y = -10.54x - 304.17 | 0.992 | -10.54C | -33.60F | -28.90–38.31 | 9.41 |
| Zhongnongjinhui | y = -8.20x - 215.98 | 0.988 | -8.20B | -32.44CDE | -26.41–38.46 | 12.05 |
| Chunmei | y = -8.03x - 207.56 | 0.975 | -8.03AB | -32.07CDE | -28.17–37.98 | 9.81 |

Note: LT-I indicated the regression line of the lethal injury temperature; R2:The coefficients of determination; Qlt indicated the slope of LT-I; LT50 indicated the half lethal temperature; LT0-LT100/˚C indicated temperature range corresponding to 0–100% injury degree severity; The data within a column followed by different big letters indicate significant difference (p<0.01).

The second exothermic process was at a lower temperature, which was directly related to freezing injury. This might have been due to the death of xylem parenchyma cells caused by deep super cooled water freezing. It was observed that the secondary exothermic process occurred in the separated xylem, and its exotherm resembled that of the branches. Freezing injury of the xylem was directly caused by the freeze of super cooled water. However, no second exotherm in the bark, suggesting that freezing damage is minimally influenced by the peach skin.

**Table 2. Computation of 16 peach varieties at different low temperatures.**

| Variety | -17˚C | -19˚C | -21˚C | -23˚C | -25˚C | -27˚C | -29˚C | -31˚C | -33˚C | -35˚C | -37˚C | -39˚C | -41˚C |
|---|---|---|---|---|---|---|---|---|---|---|---|---|---|
| Zhongtaohongyu | 0 | 0 | 0 | 0 | 4.20 | 19.50 | 34.80 | 50.10 | 65.40 | 80.70 | 96.00 | 100 | 100 |
| Qiuyan | 0 | 0 | 0 | 0 | 0 | 0 | 12.10 | 33.24 | 54.38 | 75.52 | 96.66 | 100 | 100 |
| Hunchun | 0 | 0 | 0 | 0 | 0 | 0 | 2.60 | 25.14 | 47.68 | 70.22 | 92.76 | 100 | 100 |
| 21st Century | 0 | 0 | 0 | 13.07 | 26.17 | 39.27 | 52.37 | 65.47 | 78.57 | 91.67 | 100 | 100 | 100 |
| Spring Snow | 0 | 0 | 0 | 0 | 5.38 | 20.2 | 35.02 | 49.84 | 64.66 | 79.48 | 94.3 | 100 | 100 |
| Zhonghuashoutao | 0 | 0 | 0 | 2.51 | 15.45 | 28.39 | 41.33 | 54.27 | 67.21 | 80.15 | 93.09 | 100 | 100 |
| Qiuyi | 0 | 0 | 0 | 0 | 0 | 15.12 | 30.82 | 46.52 | 62.22 | 77.92 | 93.62 | 100 | 100 |
| Golden honey No.1 | 0 | 0 | 0 | 6.41 | 19.55 | 32.69 | 45.83 | 58.97 | 72.11 | 85.25 | 98.39 | 100 | 100 |
| Okubo | 0 | 0 | 0 | 0 | 2.07 | 14.99 | 27.91 | 40.83 | 53.75 | 66.67 | 79.59 | 92.51 | 100 |
| Yanhong | 0 | 0 | 0 | 0 | 0 | 5.51 | 20.05 | 34.59 | 49.13 | 63.67 | 78.21 | 92.75 | 100 |
| Yufei | 0 | 0 | 0 | 0 | 2.09 | 18.07 | 34.05 | 50.03 | 66.01 | 81.99 | 97.97 | 100 | 100 |
| Wangjiazhuangmaotao No.2 | 0 | 0 | 0 | 0 | 0 | 0 | 0 | 11.13 | 35.75 | 60.37 | 84.99 | 100 | 100 |
| Zhongyoutao No.8 | 0 | 0 | 0 | 0 | 4.31 | 19.49 | 34.67 | 49.85 | 65.03 | 80.21 | 95.39 | 100 | 100 |
| Donghe No.1 | 0 | 0 | 0 | 0 | 0 | 0 | 1.49 | 22.57 | 43.65 | 64.73 | 85.81 | 100 | 100 |
| Zhongnongjinhui | 0 | 0 | 0 | 0 | 0 | 5.42 | 21.82 | 38.22 | 54.62 | 71.02 | 87.42 | 100 | 100 |
| Chunmei | 0 | 0 | 0 | 0 | 0 | 9.25 | 25.31 | 41.37 | 57.43 | 73.49 | 89.55 | 100 | 100 |

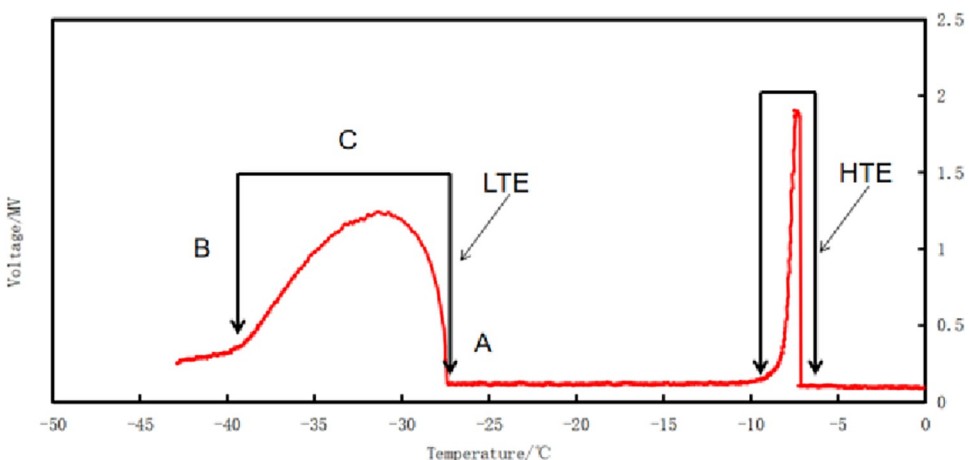

**Fig 1. Typical differential thermal analysis (DTA) profiles of low temperature exotherms (LTE) and high temperature exotherms (HTE) of peach 1a branch.**

Point A shows the initial freezing point of xylem. Point B shows the complete xylem exothermic point. Point C shows that xylem exothermic reached the peak. In Fig 1, Point A represents the initial freezing point of xylem, denoted as LT0, indicating 0% xylem injury severity. Besides, as shown in Table 1, Point B indicates the complete exothermic point of xylem, denoted as LT100, representing 100% xylem injury severity. There were significant differences in the low-temperature exothermic temperature of the different varieties. According to the data in Table 1, the ranking of the plant varieties for the LT50 of xylem was as follows: 21st Century > Golden honey No.1 > Zhonghuashoutao > Zhongtaohongyu > Yufei > Zhongyoutao No.8 = Spring snow > Qiuyi > Chunmei > Okubo > Zhongnongjinhui > Qiuyan > Yanhong > Hunchun > Donghe No.1 > Wangjiazhuangmaotao No.2. Also, it is indicated that the LT50 of the 21st Century and Golden honey No.1 varieties was higher than -30˚C, and that of the Wangjiazhuangmaotao No.2 variety was below-34˚C; the rest of the varieties had LT50 values between-30˚C and-33˚C. The difference of LT50 between the 21st Century and Wangjiazhuangmaotao No.2 varieties was 5.52˚C (Table 1).

To plot the xylem graphs (Table 1), a three-point regression analysis was investigated. The points on the graph, namely LT0, LT100, and LT50 corresponded to the initial freezing point A, final freezing exothermic B, and peak C of the xylem as depicted in Fig 1. Regression coefficient (Qlt), which is the slope of the LT-I graph, indicates the intensification of the severity of branch injury for the drop in temperature by 1˚C. With an increase in the absolute Qlt value, the varieties become more vulnerable to the drop in temperature and freezing injury. According to the data in Table 1, the ranking of the xylem Qlt of all varieties were as follows: Wangjiazhuangmaotao No.2 < Hunchun < Qiuyan < Donghe No.1 < Zhongnongjinhui < Chunmei < Yufei < Zhongtaohongyu < Zhongyoutao No.8 < Qiuyi < Spring snow < Yanhong < Golden honey No.1 < 21st Century < Zhonghuashoutao < Okubo. The correlation between Qlt and LT0 –LT100 was significantly positive (correlation coefficient of 0.918). The 1a branch temperature range of the different peach varieties was found to be broad, and the ability to resist the damage of cooling was high (Table 2).

## Comparison of cold hardiness of different peach varieties

Gao Zhen (Gao et al., 2014) conducted an estimation and verification, revealing two distinct types of LT-I in the case of root cold hardiness: intercepted and non-intercepted types, each

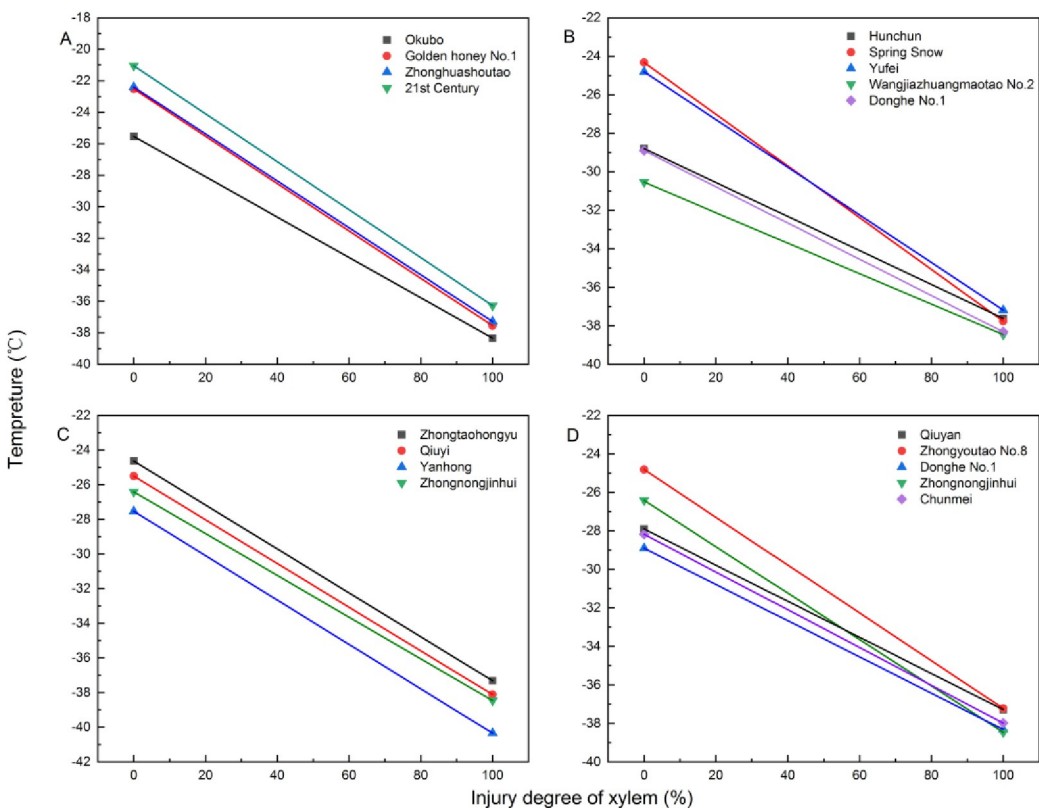

**Fig 2.** The crossing of the regression line of the lethal injury temperature (LT-I) of different peach varieties 16 cultivars were divided to four groups: A, B, C, D.

further classified into three subtypes. A similar classification was observed for the cold hardiness of branches (Fig 2).

In terms of the cold hardiness of non-intercepted varieties, three sub types were identified. For the first subtype, there is no significant difference between Qlt and LT50, which shows that the cold hardiness of the branches of the two peach varieties is basically the same. Such as: Zhongyoutao No.8 and Yufei (Table 1 and Fig 2B and 2D), Qiuyi and Yanhong (Table 1 and Fig 2C). The second type (Qlt) showed no significant difference, but LT50 exhibited some variations. For instance, Hunchun and Donghe No.1 (Table 1 and Fig 2B), Zhongtaohongyu and Qiuyi (Table 1 and Fig 2C), Zhongnongjinhui and Zhongyoutao No.8 (Table 1 and Fig 2D). The performance of the two peach varieties in LT-I were parallel. The lower the LT50, the stronger the cold hardiness was seen in the branches. The third type, Qlt and LT50 showed significant differences, the lower the LT50, the stronger the cold hardiness of the 1a branches was seen. Additionally, a smaller Qlt indicated less injury when the temperature drops by the same degree r (Table 2), such as that in 21st Century and Okubo (Table 1 and Fig 2A), Hunchun and Donghe 1 (Table 1 and Fig 2B), and Qiuyan and Donghe 1 (Table 1 and Fig 2D), all conform to this type.

Comparison of the cold hardiness of LT-I intercepted varieties was conducted. The presence of a significant difference between Qlt and LT50 was used to compare cold hardiness, among the intercepted varieties, Similar Qlt and LT50 values indicated no difference in cold hardiness between the two varieties. On the contrary, a significant difference between Qlt and LT50 indicated that cold hardiness should be assessed before and after the interception point.

Gao (2014) classified the intercepted LT-I values of roots into three subtypes depending on the LT50 occurrence relative to the interception point. For branch comparisons, there could be three subtypes as well. For the first subtype, the intersection point is before LT50. The interception point (–30.67˚C) of the LT-I of the Spring snow and Yufei varieties was noted before the LT50 (Table 1 and Fig 2B). At temperatures exceeding –30.67˚C, Yufei exhibited stronger cold-hardiness than Spring snow, while Spring snow showed better cold-hardiness below that temperature. Additionally, Qiuyan and Chunmei also belonged to this subtype (Table 1 and Fig 2D). In the second subtype, the intersection point (-34.20˚C) of Qiuyan and Zhongnong-jinhui occurred after LT50 (Table 1 and Fig 2D). Zhongnongjinhui exhibited stronger cold-hardiness above -34.20˚C, while Qiuyan showed better cold-hardiness below that temperature. For the third subtype (not displayed), the intersection point and LT50 were coincident, which did not occur in this study.

## Comparison of injury degree of different peach varieties under programmed cooling

By the LT-I regression equation, the degree of injury under different temperatures was determined (Table 1). Herein., the degree of injury to branches less than 0% was recorded as 0, and the degree of injury greater than 100% was recorded as 100 (Table 2). The degree of injury to the branches of the 21st Century, Golden honey No.1 and Zhonghuashoutao varieties were 13.07%, 6.41%, and 2.51%, respectively, at –23˚C, while that of Wangjiazhuangmaotao No.2 was 11.13% at –31˚C. The degree of injury of Qiuyan, Hunchun, and Donghe No.1 was 33.24%, 25.14%, and 22.57%, respectively, at –31˚C. It is generally believed that branch injuries exceeding 20% have an impact on production. These findings revealed that the low cold-hardiness varieties of 21st century, Golden honey No.1, and Zhonghuashoutao injuries exceeding 20% between –25˚C and –27˚C, while high cold-hardiness varieties experienced injuries exceeding 20% between –31˚C and –33˚C. Furthermore, injuries exceeding 20% was observed in the varieties between high hardiness and low hardiness between –27˚C and –29˚C. The temperature-sensitivity range of the different cold-hardiness peach varieties was different, for example, the low cold-hardiness varieties (21st Century, Golden honey No.1, Zhonghuashou-tao) showed sensitivity between –25˚C and –27˚C, the high cold-hardiness varieties showed sensitivity between –31˚C and –33˚C, and the medium cold-hardiness varieties showed sensitivity between –27˚C and –29˚C (Table 2).

When the winter temperature dropped below-33˚C, most of the varieties had frostbites or even died and the degree of injury reached 50%-70%. However, more than half of the 1a branches of high cold-hardiness varieties could safely overwinter, especially Wangjiazhuang-maotao No.2, Donghe No.1, and Hunchun, whose degrees of injury were 35.75%, 43.65%, and 47.68%, respectively (Table 2).

## Comprehensive assessment of LTE data using the subordinate function method

As shown in Table 3, the subordinate function method was used for analyzing the LTE data. Varieties with higher function values demonstrated stronger cold hardiness. Among the branches, the 21st Century, Golden honey No.1, and Zhonghuashoutao exhibited the weakest cold-hardiness (WH). While Zhongtaohongyu, Spring snow, Yufei, and Zhongyoutao No.8 had low cold-hardiness (LH). Qiuyi, Okubo, Zhongnongjinhui, and Chunmei had medium cold-hardiness (MH), Qiuyan and Yanhong had good cold-hardiness (H), and Wangjiazhuangmaotao No.2, Donghe No.1, and Hunchun had the highest cold-hardiness (HH).

**Table 3. Identification results of cold resistances of 16 peach varieties.**

| Variety | Subordination value | | | | | Cold-hardiness type |
|---|---|---|---|---|---|---|
| | Qlt | Z0/˚C | LT50/˚C | Z100/˚C | Average | |
| Zhongtaohongyu | 0.20 | 0.38 | 0.43 | 0.26 | 0.32 | LH |
| Qiuyan | 0.70 | 0.72 | 0.71 | 0.25 | 0.60 | H |
| Hunchun | 0.82 | 0.82 | 0.83 | 0.33 | 0.70 | HH |
| 21st Century | 0.02 | 0.00 | 0.00 | 0.00 | 0.00 | WH |
| Spring Snow | 0.16 | 0.34 | 0.43 | 0.37 | 0.33 | LH |
| Zhonghuashoutao | 0.00 | 0.14 | 0.31 | 0.25 | 0.18 | WH |
| Qiuyi | 0.24 | 0.47 | 0.51 | 0.45 | 0.41 | MH |
| Golden honey No.1 | 0.02 | 0.15 | 0.18 | 0.32 | 0.17 | WH |
| Okubo | 0.00 | 0.47 | 0.68 | 0.51 | 0.42 | MH |
| Yanhong | 0.14 | 0.68 | 0.81 | 1.00 | 0.66 | H |
| Yufei | 0.26 | 0.40 | 0.43 | 0.23 | 0.33 | LH |
| Wangjiazhuangmaotao No.2 | 1.00 | 1.00 | 1.00 | 0.54 | 0.88 | HH |
| Zhongyoutao No.8 | 0.19 | 0.40 | 0.43 | 0.23 | 0.31 | LH |
| Donghe No.1 | 0.70 | 0.83 | 0.90 | 0.50 | 0.73 | HH |
| Zhongnongjinhui | 0.30 | 0.56 | 0.69 | 0.54 | 0.52 | MH |
| Chunmei | 0.27 | 0.75 | 0.62 | 0.42 | 0.52 | MH |

Note: Qlt: the slope of xylem LT-I injury of 1-year-old branches; Z0: the freezing point of xylem, that is, the injury degree of xylem is 0; Z100: the xylem freezes completely, that is, the injury degree of xylem is 100%; LT50: the half lethal temperature of branches calculated according to the regression equation LT-I in Table 1.

## The relationship between the LTE subordination value of a branch and its BR and EI

Based on the subordinate function values (Table 3), three representative varieties with cold hardiness ranks of strong, moderate, and weak were selected from the studied varieties (Table 4 and Fig 3). The EI and BR of these varieties were determined, the degree of damage to the xylem was calculated, and correlation analysis was performed.

Different peach cultivars exhibited varying degrees of browning in the xylem of the 1a branch under low-temperature treatment. Wangjiazhuangmaotao No.2, Zhongnongjinhui, and Chunmei branches showed no damage at -23˚C (Table 4 and Fig 3), classified as Grade 0 according to the BR classification for freeze injury. At –27˚C, the high hardiness variety Wangjiazhuangmaotao No.2 had no change, but the medium cold-hardiness varieties (Zhongnongjinhui and Chunmei) had a xylem browning rate of 20%, and the freezing injury was Grade 1, i.e., it was slightly affected by freezing. When the freeze injury level of Wangjiazhuangmaotao No.2 reached Grade 1, the injury level of Zhongnongjinhui and Chunmei reached Grade 3 at –31˚C. When the freeze injury level of Wangjiazhuangmaotao No.2 reached Grade 3, the injury level of Zhongnongjinhui and Chunmei reached Grade 4, and the freezing injury was 100%. The results were consistent with the xylem injury data measure in LTE during the cooling process. Correlation analysis of the three peach cultivars showed a positive correlation between BR and injury of xylem, with correlation coefficients were 0.991, 0.999, and 0.907, respectively (Table 5). This indicated that the LTE analysis accurately reflected the cold-hardiness of 1a peach branches.

We analyzed the degree of injury to 1a branches of the three peach cultivars by electrolytic leakage (EI), and the results are shown in Table 4. The association of EI with the severity of xylem injury, calculated according to the LT-I and adequate values of the subordinate function (Table 4), was determined. The results showed that the degree of injury to the xylem was

**Table 4. Comparison of LTE and traditional indexes of three peach varieties.**

| Temperature /°C | (LTE) Injury degree of xylem/% | | | (Traditional indexes) Browning rate of xylem /% | | | (Traditional indexes) Injury degree determined by electrical conductivity/% | | |
|---|---|---|---|---|---|---|---|---|---|
| | Wangjiazhuangmaotao No.2 | Zhongnongjinhui | Chunmei | Wangjiazhuangmaotao No.2 | Zhongnongjinhui | Chunmei | Wangjiazhuangmaotao No.2 | Zhongnongjinhui | Chunmei |
| -23°C | 0.00 | 0.00 | 0.00 | 0.00 | 0.00 | 0.00 | 28.70 | 29.05 | 28.93 |
| -27°C | 0.00 | 5.42 | 9.25 | 0.00 | 20.00 | 26.00 | 30.56 | 31.64 | 32.26 |
| -31°C | 11.13 | 38.22 | 41.37 | 4.00 | 50.00 | 88.00 | 36.41 | 48.27 | 50.41 |
| -35°C | 60.37 | 71.02 | 73.49 | 72.00 | 96.00 | 98.00 | 59.87 | 71.24 | 72.35 |
| -39°C | 100.00 | 100.00 | 100.00 | 100.00 | 100.00 | 100.00 | 74.22 | 74.35 | 78.00 |

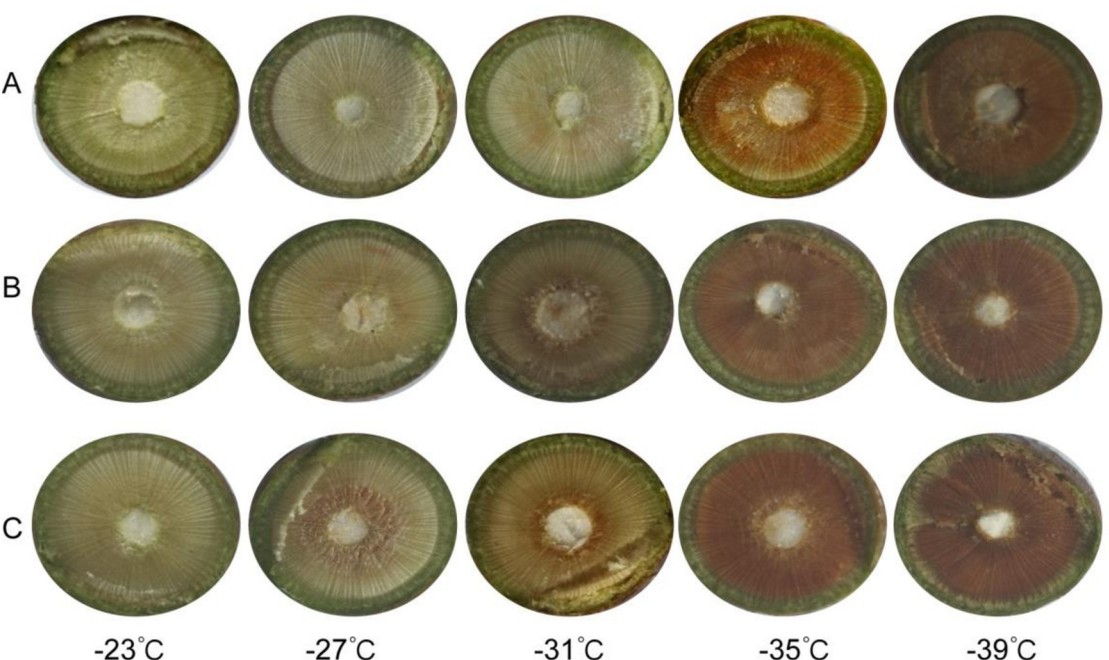

**Fig 3. Freezing injury of xylem of 1a branches of peach after low temperature treatment.** A: Wangjiazhuangmaotao No.2 B: Zhongnongjinhui C: Chunmei.

positively correlated with the EI, and the correlation coefficients of the three peach cultivars were 0.997, 0.950, and 0.991, respectively (Table 5), demonstrating that the LTE analysis accurately reflected the cold-hardiness of the 1a peach branches.

**Table 5. Correlation analysis of LTE and traditional indexes of three peach varieties.**

| Variety | Index | LTE indexes | Traditional indexes | |
|---|---|---|---|---|
| | | Injury degree of xylem | Browning rate of xylem | Injury degree determined by electrical conductivity |
| WangjiazhuangmaotaoNo.2 | Injury degree of xylem | 1 | | |
| | Browning rate of xylem | 0.991** | 1 | |
| | Injury degree determined by electrical conductivity | 0.997** | 0.993** | 1 |
| Zhongnongjinhui | Injury degree of xylem | 1 | | |
| | Browning rate of xylem | .999** | 1 | |
| | Injury degree determined by electrical conductivity | .950* | .943* | 1 |
| Chunmei | Injury degree of xylem | 1 | | |
| | Browning rate of xylem | .907* | 1 | |
| | Injury degree determined by electrical conductivity | .991** | .920* | 1 |

Note: * Significant at 0.05 levels

** Significant at 0.01 levels.

## Discussion

### Analysis of cold-hardiness of the different peach branches examined

The cold hardiness analysis was performed by the LT-I procedure for the 16 studied cultivars. Among the cultivars examined, Wangjiazhuangmaotao No.2, Donghe No.1, and Hunchun had the strongest cold-hardiness (HH), and Qiuyan and Yanhong had strong cold-hardiness (H), while 21st century, Golden honey No.1, and Zhonghuashoutao exhibited the weakest cold hardiness (WH). The inter-variety differences in cold hardiness were primarily due to the variations in the plant genetic traits. Different peach varieties, on exposure to different environments for a prolonged period, developed different genotypes, which led to variations in those traits that helped to adapt to the environment, such as cold hardiness [23]. Additionally, geologic origin and acclimatization are also factors that influence cold hardiness. For example, the cold hardiness of the northern varieties, originating from cold areas, Wangjiazhuangmaotao No.2, Donghe No.1, and Hunchun, was higher than that of the southern varieties, Golden honey No.1 and Zhongyou No.8.

### Relationship between BR and cold-hardiness branches

Tissue browning is one of the traditional methods to identify the cold hardiness of plants. Since Levitt (1956) first applied the tissue browning method to study cold-hardiness, significant progress has been made in the improvement of the method, theoretical discussion, and practical application, and it has become a fast and reliable method to study stress physiology [24]. In this study, the results of the correlation analysis of the three peach cultivars showed that the degree of injury to the xylem was positively correlated with the BR of the xylem (Table 4 and Fig 3). Similarly, Mills [21] also proved that bud and cane LTE recorded by the DTA system had a strong correlation, and the results of the severity of injury to cane xylem and phloem from tissue browning tests were similar to those based on the LTE analysis. This indicated that the LTE analysis could accurately reflect the cold-hardiness of peach branches. javascript:void(0);

### Relationship between EI and cold-hardiness of branches

One of the primary parts affected by freezing injury in plants is the cell membrane. The injury of low temperature to the membrane can lead to an increase in the EI [3]. Correlation analysis showed that the heat release of the xylem at low temperatures was associated with the permeability of the protoplasm membrane. In this study, the EI of Wangjiazhuangmaotao No.2, Zhongnongjinhui, and Chunmei were measured at different temperatures. It was found that the relative conductivity of Wangjiazhuangmaotao No.2 was lower than that of Zhongnongjinhui and Chunmei at the same temperature (Table 4), which indicated that the cold hardiness of Wangjiazhuangmaotao No.2 was higher than that of Zhongnongjinhui and Chunmei. The degree of injury to the plasma membrane of Wangjiazhuangmaotao No.2 was lower than that in the plasma membrane of Zhongnongjinhui and Chunmei, which was consistent with the degree of injury to the xylem measured in the LTE stage during programmed cooling. This indicated that the low-temperature exotherms of the xylem were related to the permeability of the plasma membrane.

### Detection method of cold-hardiness of peach branches

Cold hardiness analysis based on LTE has been extensively conducted in grape research [13, 21, 23], suggesting that it is a reliable analytical technique. In this three-point study, we performed branch LT-I determination on various peach cultivars using LT0, LT50, and LT100

through LTE investigation of cold hardiness, which was faster, simpler, and more accurate than the previous approaches.The results showed that the LTE curve of peach branches in the dormancy period had a single peak, which separated the xylem from the bark. The supercooled water of the separated xylem was similar to that of the branches. It was found that freezing injury to the xylem was directly caused by supercooled water freezing. There was no second exothermic heat in the bark, which indicated that the freezing injury to the bark was not associated with the low-temperature exothermic heat, and the results were similar to previous findings [18, 20]. In conclusion, the cold-hardiness of the 1a peach branches was mainly determined by the xylem. However, the cold-hardiness of some tree species is determined by the phloem, i.e., when the degree of injury to the phloem reaches 100%, the branches freeze to death, for example, in grape and chestnut [20, 25].

## Conclusions

In conclusion, the DTA system was utilized to perform LTE test on 1a branches of peach varieties. The application of LT-I regression linear allows for a complete and comprehensive comparison of branch cold hardiness among different varieties, accurate identifying the temperature at which xylem experience frost damage. The test method is simpler, faster and more reliable compared to determining physiological index, making it suitable for evaluating the cold hardiness of branches in different peach varieties. The comparison of LTE composite indices revealed that that Wangjiazhuangmaohuo No.2, Hunchun and Donghe No.1 belong to the high cold resistance type; Qiuyan and Yanhong are the next highest level of cold hardiness, while 21st Century, Golden Honey 1 and Zhonghuashoutao displayed the lowest cold hardiness and the least cold hardiness type among the tested varieties. In addition, the injury degrees of xylem from LT-I analysis were significantly related to the browning rates and electrolytic conductivity from traditional low temperature freezing analysis. Based on the above research results, it is believed that the LTE analysis method is a simple, accurate, and practical method for evaluating the cold resistance of peach branches, which can be used for identifying the cold hardiness of peach germplasm resources in the future.

## Acknowledgments

We are grateful for the technical support provided by Bin Han and Jicheng Han.

## Author Contributions

**Conceptualization:** Yonghong Li.

**Data curation:** Yonghong Li, Jie Li, Zhaoyuan Wang, Yu Wang, Ruifeng Chang, Hu Chen, Qihang Tian, Xiaodi Wang.

**Formal analysis:** Jie Li.

**Investigation:** Jie Li, Zhaoyuan Wang, Guojian Liu, Yu Wang, Qihang Tian, Xiaodi Wang.

**Methodology:** Yonghong Li, Jie Li, Zhaoyuan Wang, Guojian Liu, Xiaodi Wang.

**Project administration:** Yonghong Li, Xiaodi Wang.

**Resources:** Guojian Liu, Qihang Tian.

**Visualization:** Yonghong Li.

**Writing – original draft:** Yonghong Li, Jie Li, Guojian Liu, Qihang Tian, Xiaodi Wang.

**Writing – review & editing:** Yonghong Li, Guojian Liu, Qihang Tian, Xiaodi Wang.

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
