## [Decision Letter · Decision Letter 0]

14 Mar 2024

PONE-D-23-42629Analyzing Cold Hardiness (Based on DTA) of One-Year-Old Branches of PeachesPLOS ONE

Dear Dr. Li,

Thank you for submitting your manuscript to PLOS ONE. After careful consideration, we feel that it has merit but does not fully meet PLOS ONE’s publication criteria as it currently stands. Therefore, we invite you to submit a revised version of the manuscript that addresses the points raised during the review process.

We look forward to receiving your revised manuscript.

Kind regards,

Guanfu Fu, Ph.D

Academic Editor

PLOS ONE

Journal Requirements:

This work was supported by Key Laboratory of Biology and Genetic Improvement of Horticultural Crops (Germplasm Resources Utilization), Ministry of Agriculture,P.R.China (No. NYZS202305) and Technology Innovation Special Project of Hebei Academy (No. 2022KJCXZX-CGS-6), Key R&D Projects of Hebei Province (No. 21326310D), Modern Agricultural Industrial Technology System of Hebei Province (No. HBCT2023130205) and China Agriculture Research System of MOF and MARA (No. CARS-30-Z-02).

This work was supported by Key Laboratory of Biology and Genetic Improvement of Horticultural Crops (Germplasm Resources Utilization), Ministry of Agriculture,P.R.China (No. NYZS202305) and Technology Innovation Special Project of Hebei Academy (No. 2022KJCXZX-CGS-6), Key R&D Projects of Hebei Province (No. 21326310D), Modern Agricultural Industrial Technology System of Hebei Province (No. HBCT2023130205) and China Agriculture Research System of MOF and MARA (No. CARS-30-Z-02).

This work was supported by Key Laboratory of Biology and Genetic Improvement of Horticultural Crops (Germplasm Resources Utilization), Ministry of Agriculture,P.R.China (No. NYZS202305) and Technology Innovation Special Project of Hebei Academy (No. 2022KJCXZX-CGS-6), Key R&D Projects of Hebei Province (No. 21326310D), Modern Agricultural Industrial Technology System of Hebei Province (No. HBCT2023130205) and China Agriculture Research System of MOF and MARA (No. CARS-30-Z-02).

5. We note that your Data Availability Statement is currently as follows: All relevant data are within the manuscript and its Supporting Information files

7. PLOS requires an ORCID iD for the corresponding author in Editorial Manager on papers submitted after December 6th, 2016. Please ensure that you have an ORCID iD and that it is validated in Editorial Manager. To do this, go to ‘Update my Information’ (in the upper left-hand corner of the main menu), and click on the Fetch/Validate link next to the ORCID field. This will take you to the ORCID site and allow you to create a new iD or authenticate a pre-existing iD in Editorial Manager. Please see the following video for instructions on linking an ORCID iD to your Editorial Manager account: https://www.youtube.com/watch?v=_xcclfuvtxQ

Reviewers' comments:

Reviewer's Responses to Questions

**Comments to the Author**

1. Is the manuscript technically sound, and do the data support the conclusions?

Reviewer #1: Yes

Reviewer #2: Yes

2. Has the statistical analysis been performed appropriately and rigorously? 

Reviewer #1: Yes

Reviewer #2: Yes

3. Have the authors made all data underlying the findings in their manuscript fully available?

Reviewer #1: Yes

Reviewer #2: Yes

4. Is the manuscript presented in an intelligible fashion and written in standard English?

Reviewer #1: Yes

Reviewer #2: Yes

5. Review Comments to the Author

Reviewer #1: Comments

The authors selected Wangjiazhuangmaotao No.2, Zhongnongjinhui, and Chunmei as three representative varieties with strong, moderate, and weak cold hardiness ranks to determine EI and BR. However, Table 3 showed that Zhongnongjinhui and Chunmei are both medium cold-hardiness (MH) variety. In the subsequent results, there was a difference in data of BR between Zhongnongjinhui and Chunmei, both of which were MH variety. Does this indicat that the identification of cold hardness through LTE investment of is not as precise? Please explain in detail.

Minor points

1. Please review the manuscript for punctuation errors throughout the text, e.g. line 88 "... electrical leakage (EI)]".

2. Please review the manuscript for the consistency of font size, e.g. line 96 "A total of …", and line 164 "…hardiness…".

3. Please carefully check the reference format and ensure consistency, e.g. line 346 "Am. J. Enol. Vitic", and line 352 "Annals of Botany".

Reviewer #2: In this manuscript, the authors demonstrated the validity of LTE analysis for cold hardiness of peach branches by analyzing the exothermic behavior, brown rates, and the electrolytic leakage of one-year-old branches of sixteen peach cultivars under low temperature conditions. This work could be accepted after minor revisions. Other questions were shown below:

1. There are multiple formatting errors throughout the manuscript, such as the incorrect use of spaces in lines 34, 66, 74, 77, etc.

2. Line 80: "the LTE determination method involved cutting" should not be formatted in bold.

3. Fig 1: Graphing error, there is no point C in the diagram.

4. Line 140: “There” should be “there”.

5. Line 256: An “of” should be removed.

6. Line 260: “wiht” should be “with”.

7. Line 278: “siince” should be “since”.

6. PLOS authors have the option to publish the peer review history of their article (what does this mean?). If published, this will include your full peer review and any attached files.

Reviewer #1: No

Reviewer #2: No

---

## [Author Response · Author response to Decision Letter 0]

8 May 2024

Response to the comments of the academic editor and the reviewers

Manuscript Number: PONE-D-23-42629

Manuscript Title: Analyzing Cold Hardiness (Based on DTA) of One-Year-Old Branches of Peaches

Dear Editors: 

Thanks for your kind comments. We have carefully read the whole manuscript and revised it in accordance with the academic editor and the reviewers’ comments. The corrections and additions are highlighted in red color in manuscript. We list the reply as follows:

Guanfu Fu, Ph.D

Academic Editor

Reply:We have revised the manuscript according to PLOS ONE's style requirements.

Reply:

No permits statement 

We are a non-profit scientific research institutions and do not need to provide a permit.

This work was supported by Key Laboratory of Biology and Genetic Improvement of Horticultural Crops (Germplasm Resources Utilization), Ministry of Agriculture,P.R.China (No. NYZS202305) and Technology Innovation Special Project of Hebei Academy (No. 2022KJCXZX-CGS-6), Key R&D Projects of Hebei Province (No. 21326310D), Modern Agricultural Industrial Technology System of Hebei Province (No. HBCT2023130205) and China Agriculture Research System of MOF and MARA (No. CARS-30-Z-02).

Reply:We have submitted our amended Funding Statement in cover letter.

This work was supported by Key Laboratory of Biology and Genetic Improvement of Horticultural Crops (Germplasm Resources Utilization), Ministry of Agriculture,P.R.China (No. NYZS202305) and Technology Innovation Special Project of Hebei Academy (No. 2022KJCXZX-CGS-6), Key R&D Projects of Hebei Province (No. 21326310D), Modern Agricultural Industrial Technology System of Hebei Province (No. HBCT2023130205) and China Agriculture Research System of MOF and MARA (No. CARS-30-Z-02).

This work was supported by Key Laboratory of Biology and Genetic Improvement of Horticultural Crops (Germplasm Resources Utilization), Ministry of Agriculture,P.R.China (No. NYZS202305) and Technology Innovation Special Project of Hebei Academy (No. 2022KJCXZX-CGS-6), Key R&D Projects of Hebei Province (No. 21326310D), Modern Agricultural Industrial Technology System of Hebei Province (No. HBCT2023130205) and China Agriculture Research System of MOF and MARA (No. CARS-30-Z-02).

Reply:We have deleted and changed the declaration in the Acknowledgments Section of my manuscript.

5. We note that your Data Availability Statement is currently as follows: All relevant data are within the manuscript and its Supporting Information files

Reply:Our submission have contained all raw data required to replicate the results of study.

Reply:I accept your suggestion that all authors decide on a data sharing plan before acceptance.

7. PLOS requires an ORCID iD for the corresponding author in Editorial Manager on papers submitted after December 6th, 2016. Please ensure that you have an ORCID iD and that it is validated in Editorial Manager. To do this, go to ‘Update my Information’ (in the upper left-hand corner of the main menu), and click on the Fetch/Validate link next to the ORCID field. This will take you to the ORCID site and allow you to create a new iD or authenticate a pre-existing iD in Editorial Manager. Please see the following video for instructions on linking an ORCID iD to your Editorial Manager account: https://www.youtube.com/watch?v=_xcclfuvtxQ

Reply:I have ensured that I have an ORCID iD.

Reviewer #1:

1. The authors selected Wangjiazhuangmaotao No.2, Zhongnongjinhui, and Chunmei as three representative varieties with strong, moderate, and weak cold hardiness ranks to determine EI and BR. However, Table 3 showed that Zhongnongjinhui and Chunmei are both medium cold-hardiness (MH) variety. In the subsequent results, there was a difference in data of BR between Zhongnongjinhui and Chunmei, both of which were MH variety. Does this indicat that the identification of cold hardness through LTE investment of is not as precise? Please explain in detail.

Reply:The definition of medium cold-hardiness (MH) variety is relatively broad, which can also be divided into three levels: high, medium, and low, so there may be differences in the EI and BR of Zhongnongjinhui and Chunmei.

2. Minor points:Please review the manuscript for punctuation errors throughout the text, e.g. line 88 "... electrical leakage (EI)]".

Reply:We have checked and corrected the the manuscript for punctuation.

We have corrected line 88 "... electrical leakage (EI)]".

3. Minor points:Please review the manuscript for the consistency of font size, e.g. line 96 "A total of …", and line 164 "…hardiness…".

Reply:We have reviewed and corrected the manuscript for the consistency of font size.

4. Minor points:Please carefully check the reference format and ensure consistency, e.g. line 346 "Am. J. Enol. Vitic", and line 352 "Annals of Botany".

Reply:We have checked and corrected the reference format and ensure consistency,e.g. line 346 "Am. J. Enol. Vitic", and line 352 "Annals of Botany", as well as others.

Reviewer #2:

1. There are multiple formatting errors throughout the manuscript, such as the incorrect use of spaces in lines 34, 66, 74, 77, etc.

Reply:We have checked and corrected formatting errors formatting errors, such as the incorrect use of spaces in lines 34, 66, 74, 77, etc.

2. Line 80: "the LTE determination method involved cutting" should not be formatted in bold.

Reply:We have checked and corrected formatting errors formatting errors.

3.Fig 1: Graphing error, there is no point C in the diagram.

Reply:We have checked and corrected the annotation error for point C in Fig 1.

4.Line 140: “There” should be “there”.

Reply:“There” is correct writing, and we have corrected the punctuation error before it. 5. Line 256: An “of” should be removed.

Reply:We have checked and corrected the error.

5.Line 260: “wiht” should be “with”.

Reply:We have checked and corrected the error.

7. Line 278: “siince” should be “since”.

Reply:We have checked and corrected the error.

6.PLOS authors have the option to publish the peer review history of their article (what does this mean?). If published, this will include your full peer review and any attached files.

 Reply:Authors agrees to option to publish the peer review history.

---

## [Decision Letter · Decision Letter 1]

28 May 2024

PONE-D-23-42629R1Analyzing Cold Hardiness (Based on DTA) of One-Year-Old Branches of PeachesPLOS ONE

Dear Dr. Tian,

Thank you for submitting your manuscript to PLOS ONE. After careful consideration, we feel that it has merit but does not fully meet PLOS ONE’s publication criteria as it currently stands. Therefore, we invite you to submit a revised version of the manuscript that addresses the points raised during the review process.

We look forward to receiving your revised manuscript.

Kind regards,

Guanfu Fu, Ph.D

Academic Editor

PLOS ONE

Journal Requirements:

Additional Editor Comments:

According to the comments, many misspellings were still found in the manuscript. it could be accepted after careful checking.

Reviewers' comments:

Reviewer's Responses to Questions

**Comments to the Author**

1. If the authors have adequately addressed your comments raised in a previous round of review and you feel that this manuscript is now acceptable for publication, you may indicate that here to bypass the “Comments to the Author” section, enter your conflict of interest statement in the “Confidential to Editor” section, and submit your "Accept" recommendation.

Reviewer #1: (No Response)

Reviewer #3: All comments have been addressed

2. Is the manuscript technically sound, and do the data support the conclusions?

Reviewer #1: Yes

Reviewer #3: Yes

3. Has the statistical analysis been performed appropriately and rigorously? 

Reviewer #1: Yes

Reviewer #3: N/A

4. Have the authors made all data underlying the findings in their manuscript fully available?

Reviewer #1: Yes

Reviewer #3: (No Response)

5. Is the manuscript presented in an intelligible fashion and written in standard English?

Reviewer #1: No

Reviewer #3: (No Response)

6. Review Comments to the Author

**Reviewer #1:** (No Response)

**Reviewer #3:** The revised manuscript provided by Li et al. has revised and respond correctly to comments from both former reviewers.

However, there is one more comment need to be amended. I found that the authors mentioned “Huangjinmi No.1” in the Abstract section of the manuscript, but I cannot find any related information about “Huangjinmi No.1” in the other sections of the manuscript. So please double check whether this is a typo of “Golden honey No. 1” (as I have read it in the Results section but not mentioned in the Abstract section) and amend the fault. I suggest the manuscript could be accepted after this fault being fixed.

7. PLOS authors have the option to publish the peer review history of their article (what does this mean?). If published, this will include your full peer review and any attached files.

Reviewer #1: No

Reviewer #3: No

---

## [Author Response · Author response to Decision Letter 1]

22 Jun 2024

Date: June 22, 2024

Dear Editors:

 Thank you for your letter and for the academic editor and the reviewers’ comments concerning our manuscript “Analyzing Cold Hardiness (Based on DTA) of One-Year-Old Branches of Peaches”. Those comments are all valuable and very helpful for revising and improving our paper, as well as the important guiding significance to our researches. We have studied comments carefully and have made correction which we hope meet with approval. 

Academic Editor

Journal Requirements: 

Q:Please review your reference list to ensure that it is complete and correct. If you have cited papers that have been retracted, please include the rationale for doing so in the manuscript text, or remove these references and replace them with relevant current references. Any changes to the reference list should be mentioned in the rebuttal letter that accompanies your revised manuscript. If you need to cite a retracted article, indicate the article’s retracted status in the References list and also include a citation and full reference for the retraction notice.

A:We have carefully checked the reference list to ensure it is complete and correct.

Responses to Editor：

Reviewers' comments:

Responses to Reviewers：I will no longer explain the affirmative response given by the reviewers, but I will provide my answers to the No and No responses given by the reviewers.

Q:5. Is the manuscript presented in an intelligible fashion and written in standard English?

Reviewer #1: No

Reviewer #3: (No Response)

A:We have found an English native speaker with a research background to review our manuscript during revision. And if you think there is any problem, you can raise it at any time. we will look for professional organizations to improve the language

Q:6. Review Comments to the Author

Reviewer #1: (No Response)

Reviewer #3: The revised manuscript provided by Li et al. has revised and respond correctly to comments from both former reviewers.

However, there is one more comment need to be amended. I found that the authors mentioned “Huangjinmi No.1” in the Abstract section of the manuscript, but I cannot find any related information about “Huangjinmi No.1” in the other sections of the manuscript. So please double check whether this is a typo of “Golden honey No. 1” (as I have read it in the Results section but not mentioned in the Abstract section) and amend the fault. I suggest the manuscript could be accepted after this fault being fixed.

A:About Golden honey No. 1 and Huangjinmi No.1 problem mentioned by the third reviewer, we have altered to use a Golden honey No. 1. (see abstract）

Q:While revising your submission, please upload your figure files to the Preflight Analysis and Conversion Engine (PACE) digital diagnostic tool, https://pacev2.apexcovantage.com/. PACE helps ensure that figures meet PLOS requirements. To use PACE, you must first register as a user. Registration is free. Then, login and navigate to the UPLOAD tab, where you will find detailed instructions on how to use the tool. If you encounter any issues or have any questions when using PACE, please email PLOS at <a href="mailto:figures@plos.org">figures@plos.org. Please note that Supporting Information files do not need this step.

A:We have uploaded figure files to the Preflight Analysis and Conversion Engine (PACE) digital diagnostic tool.

---

## [Editor Report · Decision Letter 2]

26 Jun 2024

Analyzing Cold Hardiness (Based on DTA) of One-Year-Old Branches of Peaches

PONE-D-23-42629R2

Dear Dr. Qihang Tian

We’re pleased to inform you that your manuscript has been judged scientifically suitable for publication and will be formally accepted for publication once it meets all outstanding technical requirements.

Kind regards,

Guanfu Fu, Ph.D

Academic Editor

PLOS ONE
---

## [Editor Report · Acceptance letter]

28 Jun 2024

PONE-D-23-42629R2 

PLOS ONE

Dear Dr. Tian, 

I'm pleased to inform you that your manuscript has been deemed suitable for publication in PLOS ONE. Congratulations! Your manuscript is now being handed over to our production team.

Kind regards, 

on behalf of

prof. Guanfu Fu 

Academic Editor

PLOS ONE